# The Role of Nasal Cytology and Serum Atopic Biomarkers in Paediatric Rhinitis

**DOI:** 10.3390/diagnostics13030555

**Published:** 2023-02-02

**Authors:** Giulia Dodi, Paola Di Filippo, Francesca Ciarelli, Annamaria Porreca, Fiorella Cazzato, Lorena Matonti, Sabrina Di Pillo, Giampiero Neri, Francesco Chiarelli, Marina Attanasi

**Affiliations:** 1Pediatric Allergy and Pulmonology Unit, Department of Pediatrics, University of Chieti-Pescara, 66100 Chieti, Italy; 2Department of Economic Studies, University of Chieti-Pescara, 66100 Chieti, Italy; 3Department of Otolaryngology, University of Chieti-Pescara, 66100 Chieti, Italy

**Keywords:** local allergic rhinitis, cytology, nasal provocation test, eosinophilia, serum IgE

## Abstract

A Nasal Provocation Test allows the differentiation of allergic and non-allergic rhinitis, but it is difficult and expensive. Therefore, nasal cytology is taking hold as an alternative. We carried out a cross-sectional study, including 29 patients with persistent rhinitis according to ARIA definition and negative skin prick tests. Nasal symptoms were scored from 0 to 5 using a visual analogue scale, and patients underwent blood tests to investigate blood cell count (particularly eosinophilia and basophilia), to analyze serum total and specific IgE and eosinophil cationic protein (ECP), and to perform nasal cytology. We performed a univariate logistical analysis to evaluate the association between total serum IgE, serum eosinophilia, basophils, and ECP and the presence of eosinophils in the nasal mucosa, and a multivariate logistic model in order to weight the single variable on the presence of eosinophils to level of the nasal mucosa. A statistically significant association between serum total IgE levels and the severity of nasal eosinophilic inflammation was found (confidence interval C.I. 1.08–4.65, odds ratio OR 2.24, *p* value 0.03). For this reason, we imagine a therapeutic trial with nasal steroids and oral antihistamines in patients with suspected LAR and increased total IgE levels, reserving nasal cytology and NPT to non-responders to the first-line therapy.

## 1. Introduction

Chronic rhinitis is a widespread pediatric condition characterized by rhinorrhea, nasal obstruction, epiphora, and nasal itching. It is not a life-threatening disease, and it strictly affects children’s quality of life [1,2]. The diagnosis requires inflammation of the nasal epithelium and at least two of the aforementioned symptoms [3,4].

Two subgroups of rhinitis are identified: allergic rhinitis (AR) and non-allergic rhinitis (NAR) [5]. AR is an atopic disease and depends on immunoglobulin E (IgE) sensitization to indoor and outdoor aeroallergens [6,7]. It is characterized by nasal eosinophilic inflammation [8,9], and the diagnosis requires positive skin prick tests (SPT) and/or serum specific IgE levels (sIgE). NAR is a highly heterogeneous condition, including drug-induced rhinitis, occupational rhinitis, hormonal rhinitis, gustatory rhinitis, senile rhinitis, and idiopathic rhinitis. It is due to inflammatory and noninflammatory etiologies [2,7] without IgE mediation, as documented by negative SPT and sIgE [7,10].

Atopic markers could be positive in asymptomatic subjects and negative in subjects with clinical rhinitis. However, their negativity does not necessarily exclude AR, reflecting the complexity of this disease [11]. Nasal Provocation Test (NPT) links rhinitis symptoms to allergen exposure [5]. It results negative in NAR and positive for at least one aeroallergen in AR.

Local Allergic Rhinitis (LAR) is characterized by a Th2 inflammatory response with a local nasal production of total and specific IgE. Subjects with LAR present negative atopic markers and a positive response to NTP [12]. During NPT, an allergen extract or other provocative agents are instilled directly onto the nasal mucosa and the intensity of nasal symptoms (itching, sneezing, rhinorrhoea, and nasal obstruction) is recorded using scoring systems [13,14]. Therefore, NPT represents the gold standard test to diagnose LAR [13,14].

IgE dosage in nasal lavage fluid is also a useful diagnostic tool. For example, Colavita et al. [15] evaluated the nasal lavage fluid IgE as a biomarker of LAR in children performing nasal lavage using 2 mL/nostril of physiologic saline solution, which was then analyzed by ImmunoCAP to obtain the IgE concentration. They found a higher dosage of IgE in nasal lavage fluid in 16 out of 26 patients of the study group who were classified as affected by LAR. The authors also found a better response to allergic rhinitis therapy in the LAR group than in the NAR group.

Unfortunately, neither NPT nor nasal specific IgE dosage are readily available in clinical practice [16].

For this reason, nasal cytology is considered a simple, effective, and inexpensive tool to better differentiate rhinitis phenotypes. It is performed with a nasal scraping followed by May–Grunwald–Giemsa staining and optical microscopy reading [16,17]. Nasal mucosa is physiologically constituted by ciliata, mucipara, striata, and basalis cells; other cells are not represented, except rarely neutrophils and bacteria [18,19]. Since AR is associated with nasal smear eosinophilia, nasal cytology could differentiate rhinitis with and without nasal eosinophilia [8,9]. In addition, nasal cytology distinguishes inflammatory from non-inflammatory rhinitis, identifying and counting cell types [18,20,21,22]. According to the predominant cell type, various entities can be identified: non-allergic rhinitis with eosinophils (NARES), non-allergic rhinitis with eosinophils and mast cells (NARESMA), non-allergic rhinitis with mast cells (NARMA), and non-allergic rhinitis with neutrophils (NARNE) [20,21,22].

However, nasal cytology fails to differentiate LAR from NARES since both present nasal eosinophilia [22]. Nevertheless, a first line therapy of both LAR and NARES consists of oral antihistamine and topic steroids [23].

The aim of this study was to analyze the results of nasal cytology in a group of children affected by symptoms suggestive of AR without systemic atopic markers (SPT and sIgE). We also hypothesized that serum biomarkers (total sIgE, serum eosinophilia and eosinophil cationic protein, serum basophilia) could predict nasal eosinophilia in order to simplify our daily practice and reserve more time-consuming and expensive test to selected patients.

## 2. Materials and Methods

This cross-sectional study was carried out at the Pediatric Allergy and Respiratory Unit of the University of Chieti from March 2018 to November 2019.

Preschool and school-aged children with persistent rhinitis according to Allergic Rhinitis and its Impact on Asthma (ARIA) definition (symptoms for more than 4 days/week and lasting more than 4 weeks) and negative skin prick tests were enrolled [10]. SPT for the most common food and respiratory allergens (egg white and yolk, milk, cod, tomato, wheat, peanut, shrimp, parietaria grass, olive, cypress, absinthe, ambrosia, dermatophagoides pteronyssinus and farinae, cat and dog dander, alternaria alternate) are usually performed in our unit. During the first visit, an accurate family and personal medical history was collected by a pediatric pulmonologist, including information about the family history of asthma and allergy. Anthropometric parameters were recorded and questionnaires for the assessment of nasal symptoms, impact on the quality of life, and associated comorbidities were administered according to ARIA indications [10].

Using a visual analogue scale (VAS), nasal symptoms were scored from 0 to 5 (0 = no symptoms to 5 = very severe symptoms) for five different domains: itchy eyes, itchy nose, rhinorrhea, sneezing, and nasal obstruction. The total symptom score was calculated by adding up all the symptoms (scoring range between 0 and 25) [16].

The total marks registered were classified as follows:−mild symptoms: overall score from 0 to 5,−moderate symptoms: overall score from 6 to 15,−severe symptoms: overall score from 16 to 25.

The impact on quality of life was assessed by the physician according to ARIA guidelines, investigating the influence of rhinitis on school activities, daily activities, sleep, and the perception of symptoms as troublesome. Rhinitis was distinguished as mild if none of the mentioned aspects were present and either moderate to severe if one or more of the mentioned aspects are present [10].

We also collected data about comorbidities, like conjunctivitis, sinusitis, obstructive sleep apnea syndrome, and asthma [24]. Lastly, we recorded the current therapy: nasal steroids, antihistamines, and association of nasal steroids or antihistamines.

Patients underwent blood tests to investigate blood cell count (particularly eosinophilia and basophilia), serum total and specific IgE, and eosinophil cationic protein (ECP). Total IgE were considered normal until 80 kU/L between 1 and 5 years of age or 120 kU/L over 14 years of age [25]. ECP value was considered normal until the cut off of 15 mcg/L. Eosinophilia was determined by eosinophils that were more than 5% of leukocytes while basophils that were more than 1% of leukocytes were defined as basophilia [26].

Patients were excluded if they presented (1) positive specific IgE, (2) acute respiratory infections in the last four weeks, (3) systemic corticosteroid therapy within 4 weeks, or (4) nasal steroid therapy and antihistamine within 2 weeks to avoid the understatement of the VAS scale and effect of therapy on nasal cells.

Systemic allergy was excluded by specific serum IgE for several allergens: egg white and yolk, milk, cod, wheat, peanut, soya beans, hazelnut, tomato, dermatophagoides pteronyssimus, loglierella, jewish peerage, aspergillus fumigatus, alternate alternaria, olive tree, cat and dog dander, and cupressaceae.

During the second visit, the nasal cytology samples were collected by scraping the medial surface of the middle part of inferior turbinate with a Rhinoprobe for each nostril at the Otorhinolaryngology Department. The sample of each nostril was transferred onto a labelled glass slide (right or left nostril). The slides were air-dried and stained with Wright-Giemsa method. Nasal cytology analysis was performed using an optical 1000 magnification binocular microscopy with oil immersion.

The cytologic report included a qualitative description of the sample (epithelial ciliated cells, mucinous cells, neutrophil and eosinophil, basophilic and eosinophil/mast cells degranulation) and quantitative count of inflammatory cells expressed as a percentage of the total amount. Nasal eosinophilia was established if eosinophils are present in the nasal mucosa. It was defined mild if the rate of eosinophils was 1–5%, moderate if it was between 5–20%, and severe if it was greater than 20% [13].

The study was approved by the Ethical Committee of University of Chieti, and written consent was obtained from the parents of the enrolled children.

## 3. Statistical Analysis

Continuous data were expressed as mean and standard deviation (SD) or median and range 5–95%. Categorical data were presented as numbers and percentages.

A univariate logistical analysis was performed to evaluate the association between atopic serum markers (total serum IgE, serum eosinophilia, basophils, and ECP) and the presence of eosinophils in the nasal mucosa. A multivariable logistic regression analysis was performed to examine the influence of the single variable on the primary outcome (eosinophils at the level of the nasal mucosa).

The statistical significance level was *p* < 0.05. SPSS version 25.0 for Windows (IBM, Armonk, NY, USA) and STATA/IC 15.1 (StataCorp. 2017. Stata Statistical Software: Release 15. StataCorp LLC. College Station, TX, USA) were used to perform statistical analyses.

## 4. Results

Patients with symptoms of AR and negative atopic markers were 70; 41 patients were excluded, thus 29 patients underwent nasal cytology and were included in the analysis (flowchart in Figure 1).

All participants were Caucasian and aged from 3 to 8 years. The main anthropometric and clinical characteristics of the study population are reported in Table 1. Family history of allergy and asthma was found in 16/29 (55.2%) and in 3/29 (10.3%) subjects, respectively. In 14 subjects (48.3%), one or more comorbidities were found: 8 patients (27.6%) complained of conjunctivitis, 8 (27.6%) of sinusitis, 2 (6.9%) of obstructive sleep apnoea syndrome (OSAS), and 2 (6.9%) of asthma.

Symptoms of rhinitis as perennial were referred by 12/29 (41.4%) patients. Most of patients (75.8%) presented watery rhinorrhea; 17.2% of patients did not need any therapy, whereas 27.6% of patients needed oral antihistaminic therapy, 10.3% of patients needed local steroidal therapy, and 44.9% of patients needed both.

The most bothersome symptom was nasal obstruction (20 patients of 26 presented it, i.e., 76%). The total score of VAS showed that 10 patients (10.3%) presented mild symptoms, 17 (58.6%) moderate symptoms, and only 2 (6.9%) severe symptoms. According to ARIA classification, 20.7% of patients suffered from mild rhinitis, while 79.3% of patients suffered from moderate–severe rhinitis.

We found hypereosinophilia in 2/29 patients (6.9%), increased total IgE values in 8/29 patients (27.9%), and increased ECP values in 26/29 patients (86.9%). Basophilia was not found in any patient. (Table 2)

Nasal cytology detected eosinophils in 10/29 patients (34.5%). Eosinophils represented 1 to 5% of the total cells in 4 patients, 5 to 20% of the total cells in 5 patients, and more than 20% of the total cells in 1 patient. The remainder of the reports were characterized by the presence of neutrophils and bacterial biofilm. (Table 2)

A statistically significant association between serum total IgE levels and the severity of nasal eosinophilic inflammation was found (confidence interval C.I. 1.08–4.65, odds ratio O.R. 2.24, *p* value 0.03). No association between ECP levels, serum eosinophils and basophils, and nasal eosinophilic inflammation was found (C.I. 0.28–3.67, O.R. 1.01, *p* value 0.99; C.I. 0.57–3.22, O.R. 1.35, *p* value 0.49; C.I. 0.41–3.09, O.R. 1.13, *p* value 0.81) Neither VAS score was statistically associated with nasal eosinophilic inflammation (C.I. 0.81–1.16, O.R. 0.96, *p* value 0.65) (Table 3).

## 5. Discussion

In our study population, the most bothersome symptom reported was nasal obstruction. Similarly, in a survey including 500 children from 4 to 17 years of age, the single most frequently experienced symptom reported was nasal congestion or a stuffy nose (52%) [27]. Conversely, in a recent multicentre study with 253 children aged 6–11 years and 250 adolescents, ocular symptoms were the most frequently reported and most impacting on quality of life. Nasal congestion resulted in the second most reported symptom by children, but its impact on quality of life was greater in the 250 included adolescents than in children [28].

According to the VAS score, 10 patients (10.3%) presented mild symptoms, 17 (58.6%) moderate symptoms, and only 2 (6.9%) severe symptoms. According to ARIA classification, 20.7% of patients suffered from mild rhinitis while 79.3% patients suffered from moderate–severe rhinitis. Similarly, in an observational multicentre study with 806 patients, 83.5% of subjects suffered from moderate-to-severe persistent AR according to ARIA classification [28]. However, the study population included 37.6% of adults, but the authors reported the incidence of moderate-to-severe rhinitis in age groups and found an incidence of 81.2% in children aged 6–11 years and 79.6% in adolescents [28]. In an observational, cross-sectional and multicentre study including 1275 children aged 6–12 years and with AR, Jáuregui et al. [29] found that 89.7% of patients suffered from the moderate/severe type of disease.

In our study population, nasal cytology showed that 34.5% of patients with persistent rhinitis and negative SPT and sIgE presented nasal eosinophilic inflammation. In clinical practice, the diagnosis of AR in a child is indirectly inferred by medical history and IgE or SPT positivity. LAR assumes the presence of a local allergic reaction characterized by increased production of IgE and eosinophils [15]. To date, the diagnosis of LAR is based on NPT and/or the demonstration of synthesis of IgE in the nasal mucosa. The gold standard for the diagnosis of both LAR and AR is NPT [30], but it is time-consuming, difficult to perform in children, and not available in many centers. Mierzejewska et al. [8] stated that eosinophils in the nasal mucosa allow to differentiate children with and without atopy. Indeed, nasal cytology can directly detect the allergic etiology by the presence of eosinophils through the microscopy examination of the inferior turbinate cells [2,18].

In order to reduce the diagnostic cost related to NPT, we suggested that nasal cytology analysis could be used in NAR patients as a screening tool for the diagnosis of LAR. Successively, patients with eosinophil nasal inflammation should perform NPT. Contrarily, using NPT as a first approach in patients with suspected LAR, 65.5% of our patients would presumably have been unnecessarily tested. Similarly, Phothijindakul et al. [16] suggested nasal cytology as a screening tool: patients with nasal eosinophilia assessed by nasal cytology should undergo NPT. Mierzejewska et al. [8] showed that NPT should be considered in patients with nasal eosinophilia to identify LAR patients due to the low specificity of nasal cytology (Figure 2).

Nevertheless, nasal cytology cannot differentiate LAR from NARES because they are both characterized by nasal eosinophilic inflammation [13,23]. No recognized nasal eosinophilic threshold for the diagnosis of NARES was established, ranging from 5% to 20% eosinophils [23,31,32,33,34,35]. Several studies suggested that NARES mostly overlaps with other conditions. It was often considered a forerunner of aspirin triad (eosinophilic nasal polyps, non-allergic asthma, intolerance to aspirin) [36,37], a form of idiopathic rhinitis [34,37], a local inflammatory response induced by irritants [38], and even misdiagnosed local AR [11]. However, the pathogenic role of severe eosinophilic inflammation in NARES is recognized [36,37,39]. Meng et al. [40] showed that patients with NARES had higher levels of ECP and leukotriene C4 in nasal secretions, confirming the usefulness of therapy acting against eosinophilic inflammation. Therefore, therapeutic options for classical AR [41], LAR [42,43], and NARES [44,45,46,47] are intranasal corticosteroids with or without antihistamines and/or leukotriene antagonists. Therapeutic efficacy in all three types of rhinitis is due to the common eosinophilic inflammation and reactivity to allergens [40].

Finally, we found increased total IgE levels in 27.9% of patients and a statistically significant association between total IgE levels and eosinophils in nasal mucosa. It is well recognized that increased total serum IgE values suggested an atopic status [48,49]. Hu et al. [50] examined 396 patients with atopic dermatitis and found that elevated serum total IgE level, peripheral eosinophils, and basophils were more frequent in patients with atopic dermatitis than in controls (*p* < 0.05). Particularly, 62.6% of patients with AD showed elevated sIgE levels. Similarly, Sharma et al. [51] recruited 480 asthmatics/allergic patients, 100 first-degree relatives of asthmatics, and 120 unrelated normal healthy volunteers. The authors found the highest IgE levels in asthmatic patients and the lowest ones in the healthy group. Although several studies have been conducted on children to establish the reference values of serum total IgE levels, the reliability of total IgE as a diagnostic criterion of allergic diseases is still under debate [50,51,52].

In order to establish reference values of total IgE in Asian children and to assess their significance in the diagnosis of atopy and allergic diseases, Tu et al. [52] evaluated 1321 Asian children aged 5–18 years. Their multivariate analysis revealed that atopy was the single most important determinant, explaining 66.1% of the variability of total IgE levels in the study population. The authors found that sensitivity, specificity, and positive and negative predictive values of total IgE at the optimal cut-off of 77.7 kU/L on the ROC curve for diagnosing atopy were 82.3%, 87.1%, 89.5%, and 78.6%, respectively. They concluded that total IgE at the cut-off of 77.7 kU/L had high negative predictive values (84.2–97.9%) for diagnosing allergic disease.

Total serum IgE levels result from spontaneously produced IgE and IgE generated during atopic responses to environmental allergens [49]. In atopic children, total IgE levels increase when an allergen-specific IgE response initiates. In non-atopic children, total IgE levels increase slightly and very slowly during the first decade of life, remaining within normal limits [48,53]. Beyond eosinophils, a local IgE production restricted to nasal mucosa was demonstrated in patients affected by rhinitis with and without systemic atopy [15,54]. The association between systemic IgE and nasal eosinophils, which are both typical of an allergic condition, could reflect the simultaneous increase of local IgE in patients with nasal eosinophilic inflammation. Unfortunately, local IgE levels were not measured in our patients, but a recent study found that total and specific IgE in the nasal mucosa of 11 patients with AR correlated significantly with their respective serum levels [55]. An association between increased levels of total IgE, specific IgE, and eosinophilic inflammation was found in nasal polyps tissues of 20 patients [56]. Another recent study with 76 asthmatic adults showed a correlation between systemic total IgE and FeNO, which is a local airways eosinophilic inflammatory marker [57].

Therefore, the estimation of serum IgE levels could be a simple, cost-sparing, and reliable tool in the diagnostic work-up of AR [58,59]. However, most of the aforementioned studies were conducted on small samples and these findings should be studied in larger populations. A study with the simultaneous evaluation of both systemic and local total IgE and eosinophils would be desirable, to better investigate their association.

Eosinophilia is another hallmark of allergy [60]. Demirjian et al. [61] and Brakhas et al. [62] found increased serum IgE and eosinophil levels in patients with AR, indicating an atopic etiology. In our study, no association between peripheral eosinophilia and nasal eosinophils was found. In the literature, this association was found in adults with AR and nasal polyposis. Blood eosinophilia resulted in a correlation to tissue eosinophilia in patients with nasal polyps, but the correlation was moderate, suggesting a discordance between systemic, and local eosinophilic inflammation in some patients [63,64,65]. Wang et al. [66] found that blood and tissue eosinophilia were concordant in 31.2% adults with AR and nasal polyposis.

However, most studies focused on the correlation of local eosinophils with the disease phenotype, and little is known about the relationship between systemic and local eosinophilia. Nevertheless, the gene expression analysis of nasal and blood samples of AR sufferers revealed two distinct profiles, mirroring the different cellular composition and biological roles of circulating blood and the nasal mucosa [67].

We found no association between nasal eosinophilic inflammation and symptoms. Similarly, Colavita et al. [15] found no association between the severity of rhinitic symptoms and the nasal lavage fluid IgE concentration in children aged between 4 and 12 years. The lack of association could be due to an underestimation of symptoms in children. Indeed, Occasi et al. [68] investigated the correlation between rhinomanometry and nasal obstruction in 284 children aged between 6 and 14 years with AR. The authors showed that primary school children (6–9 years of age) often underestimate their nasal symptoms, concluding that an objective measurement of nasal symptoms (rhinomanometry) should be performed [68].

Concluding, since oral antihistamines and topical steroids are the first line of treatment for both LAR and NARES [42,69], we suggested a therapeutic trial with oral antihistamines and topical steroids for patients with persistent symptoms of rhinitis, negative SPT and sIgE, but increased total IgE level, which seems to be associated with nasal eosinophilia. Non-responder patients could be selected for further diagnostic evaluation (NPT and nasal cytology), avoiding second and third level tests in patients who benefit from the first line of therapy.

The major strength of this study is the comprehensive atopic evaluation of participants through the simultaneous assessment of nasal eosinophilic inflammation and systemic markers of atopy. In addition to the systemic atopy evaluation, it is important to consider the local nasal mucosal system in children with rhinitis to better differentiate the various patterns of rhinitis and evaluate the response to therapy. Finally, nasal cytology was performed by the same operator with expertise in the otolaryngology field.

However, several limitations need to be discussed. First, the small sample size could have affected the power of the study. Similarly, many studies on nasal cytology in children with AR include a small number of participants. Second, the comparison of nasal cytology results between children with AR and healthy subjects was not performed because of the absence of a control group. Third, NPT was not performed, denying the comparison between nasal cytology and NPT findings and the differentiation between LAR and NARES. In addition, an objective measurement of nasal symptoms would have been useful to assess the relationship between nasal eosinophilic inflammation and atopy systemic markers with nasal symptoms, often underestimated in children. Lastly, the relationship between total serum IgE and local IgE in the nasal mucosa is only conceivable, as local IgE level was not investigated. Further and larger studies are needed to investigate the association between nasal and systemic atopic markers.

## 6. Conclusions

With all the aforementioned limitations, we found an association between total IgE levels and nasal eosinophilia. These findings could simplify the management of children with symptoms suggestive of AR with negative markers of systemic atopy. We propose a therapeutic trial with nasal steroids and oral antihistamines in the case of increased total IgE levels, reserving further diagnostic tools (nasal cytology and NPT) to non-responders to the first line of therapy.

## Figures and Tables

**Figure 1 diagnostics-13-00555-f001:**
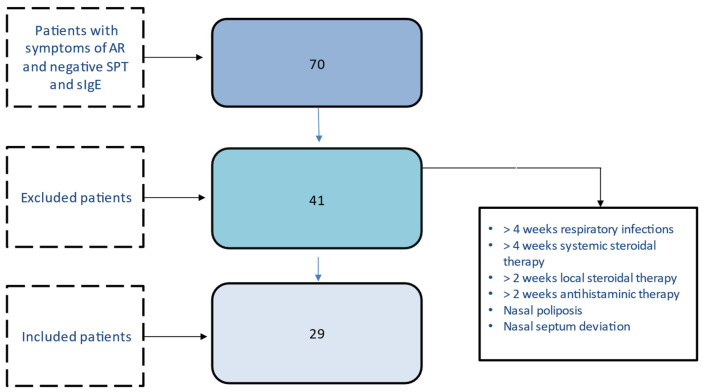
Flow-chart of the study.

**Figure 2 diagnostics-13-00555-f002:**
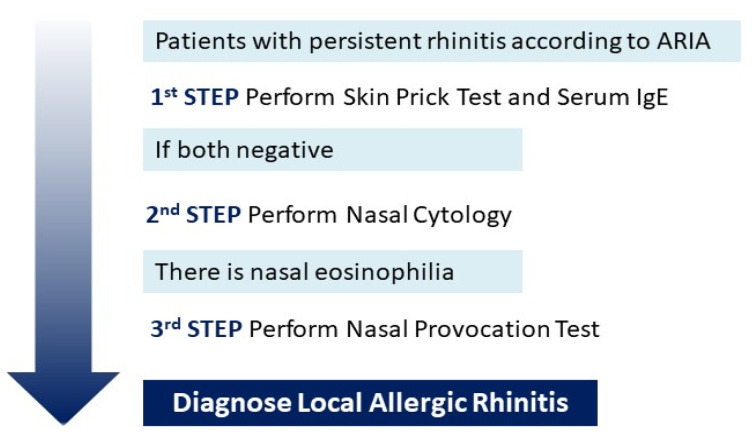
We suggest that nasal cytology analysis could be used as a screening tool for the diagnosis of LAR for those patients with persistent rhinitis according to ARIA and negative skin prick test and serum IgE. Successively, patients with eosinophil nasal inflammation should perform NPT to confirm LAR. *LAR* Local Allrergic Rhinitis; *NPT* Nasal Provocation Test.

**Table 1 diagnostics-13-00555-t001:** Anthropometric characteristics and comorbidities of the study population.

	*n* = 29
Male sex (*n*, %)	15 (51.7)
Age (years)	8.4 ± 2.8
Height (cm)	129.2 ± 15.9
Weight (kg)	29.4 ± 10.9
Family history of atopy (*n*, %)	16 (55.2)
Family history of asthma (*n*, %)	3 (10.3)
Conjunctivitis (*n*, %)	8 (27.6)
Sinusitis (*n*, %)	8 (27.6)
OSAS (*n*, %)	2 (6.9)

Data are presented as mean ± SD or *n* (%). OSAS Obstructive Sleep Apnoea Syndrome.

**Table 2 diagnostics-13-00555-t002:** Serum and nasal cytology results of the study population.

	*n* = 29
**Serum results**	
Eosinophilia (*n*, %)	2 (6.9)
Increased total IgE values (*n*, %)	8 (27.9)
ECP values (*n*, %)	25 (86.9)
Basophilia (*n*, %)	0 (0)
Cytology results	
Nasal eosinophils (*n*, %)	10 (34.5)
Mild (*n*, %)	4 (40.0)
Moderate (*n*, %)	5 (50.0)
Severe (*n*, %)	1 (10.0)
Nasal neutrophils & Bacterial biofilm (*n*, %)	19 (65.5)

Data are presented as *n* (%). ECP eosinophilic cationic protein; mild 1–5% of eosinophils, moderate 5–20% of eosinophils, severe more than 20% of eosinophils in nasal smear.

**Table 3 diagnostics-13-00555-t003:** Association of nasal eosinophilia with markers of atopy (serum eosinophils, serum basophils, ECP) and Clinical Score. ECP eosinophilic cationic protein.

	OR	CI	*p*-Value
Total IgE	2.24	1.08–4.65	**0.031**
Serum eosinophils	1.35	0.57–3.22	0.495
Serum basophils	1.13	0.41–3.09	0.813
ECP	1.01	0.28–3.67	0.991
Clinical Score	0.96	0.81–1.16	0.651

## Data Availability

The data presented in this study are available on request from the corresponding author. The data are not publicly available due to privacy reasons.

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
