# Peer review of "The Role of Nasal Cytology and Serum Atopic Biomarkers in Paediatric Rhinitis"

_diagnostics, 2023, doi:10.3390/diagnostics13030555_

Round 1

Reviewer 1 Report

1.The authors present their hypotheses and data clearly.

2. Revise line 46 to clarify the reference to sIgE in relationship to local nasal mucosa.

3. Identify literature describing background sIgE levels as a surrogate for absence of a control population.

4. Provide data to explain exclusion of patients using nasal steroids and antihistamines as clinically patients may not be cooperative with a "washout" period prior to testing.

5. Provide description of allergens included in SPT and specific serum IgE

Author Response

1. We thank the reviewer for his/her comments. 

2. Revise line 46 to clarify the reference to sIgE in relationship to local nasal mucosa.
We thank the reviewer for this useful advice. We tried to explain this content better in lines 45-57. 

3. Identify literature describing background sIgE levels as a surrogate for absence of a control population.
We thank the reviewer for his/her suggestions. We reported that the absence of a control group was a study limitation in the discussion; in agreement with the reviewer, we added studies investigating total serum IgE in atopic and non-atopic population [lines 254-274]. 

4. Provide data to explain exclusion of patients using nasal steroids and antihistamines as clinically patients may not be cooperative with a "washout" period prior to testing.
We decided to exclude patients using nasal steroids and antihistamines in order to avoid understatement of VAS scale and effect of therapy on nasal cells. Anyway, none of our excluded patients' parents agreed to stop therapy in order to include their children in the study. 

5. Provide description of allergens included in SPT and specific serum IgE
We thank the reviewer for his/her suggestion; we reported a description of allergen included in SPT and specific serum IgE in lines 87-91 and 122-125, respectively. 

Reviewer 2 Report

Currently, allergic rhinitis in children presents many diagnostic and therapeutic challenges. The current study tries to define the role of nasal cytology and blood biomarkers in the differential diagnosis of pediatric rhinitis. The study methods are well and clearly presented, inclusion criteria supported, statistical methods properly used. The originality of the study is conferred by the demonstration of a significant correlation between serum total IgE levels and severity of nasal eosinophilic inflammation. Although the authors use routine tests from clinical practice, the result is a paradigm shift in the diagnosis of pediatric allergic rhinitis. A positive diagnosis allows effective treatment, a detail rightly emphasized by the authors. In conclusion, the authors must be congratulated for the very good article that is part of the current trend of personalized medicine.

Author Response

We thank the reviewer very much for the comments on our work. We believe he/she fully understood the significance of this study in our clinical practice. 
